# A Review of Compartmentalised Inflammation and Tertiary Lymphoid Structures in the Pathophysiology of Multiple Sclerosis

**DOI:** 10.3390/biomedicines10102604

**Published:** 2022-10-17

**Authors:** Rachael Kee, Michelle Naughton, Gavin V. McDonnell, Owain W. Howell, Denise C. Fitzgerald

**Affiliations:** 1Wellcome-Wolfson Institute for Experimental Medicine, Queen’s University Belfast, Belfast BT9 7BL, UK; 2Department of Neurology, Royal Victoria Hospital, Belfast BT12 6BA, UK; 3Institute of Life Sciences, Swansea University, Wales SA2 8QA, UK

**Keywords:** multiple sclerosis, progressive multiple sclerosis, nervous system, compartmentalised inflammation, meningeal inflammation, tertiary lymphoid structures, disease modifying therapy

## Abstract

Multiple sclerosis (MS) is a chronic, immune-mediated, demyelinating disease of the central nervous system (CNS). The most common form of MS is a relapsing–remitting disease characterised by acute episodes of demyelination associated with the breakdown of the blood–brain barrier (BBB). In the relapsing–remitting phase there is often relative recovery (remission) from relapses characterised clinically by complete or partial resolution of neurological symptoms. In the later and progressive stages of the disease process, accrual of neurological disability occurs in a pathological process independent of acute episodes of demyelination and is accompanied by a trapped or compartmentalised inflammatory response, most notable in the connective tissue spaces of the vasculature and leptomeninges occurring behind an intact BBB. This review focuses on compartmentalised inflammation in MS and in particular, what we know about meningeal tertiary lymphoid structures (TLS; also called B cell follicles) which are organised clusters of immune cells, associated with more severe and progressive forms of MS. Meningeal inflammation and TLS could represent an important fluid or imaging marker of disease activity, whose therapeutic abrogation might be necessary to stop the most severe outcomes of disease.

## 1. Introduction

Multiple sclerosis (MS) is a chronic, immune-mediated, demyelinating disease of the central nervous system (CNS) [1]. Relapsing-remitting MS (RRMS) is the most common form of the disease. Patients with RRMS experience episodic relapses, characterised by neurological signs and symptoms which wax and wane. Relapses are caused by acute inflammatory demyelinating attacks in the CNS after which follows a period of remission where there is either incomplete or complete clinical recovery. A significant proportion of people with RRMS will go on to develop secondary progressive MS (SPMS) that is characterised by accumulation of irreversible disability [1,2,3,4]. A smaller number of patients are diagnosed with primary progressive MS (PPMS) characterised by a continual accrual of disability from symptom onset. In the progressive disease stages, accrual of neurological disability occurs in a pathological process independent of relapses and is accompanied by a trapped or compartmentalised inflammatory response, most notable in the connective tissue spaces of the vasculature and leptomeninges. Recent studies suggest that the incidence of MS is increasing [5,6]. Most people with MS are diagnosed between 20–40 years of age and so this demyelinating disease has huge personal and economic costs [7]. Whilst disease-modifying treatments (DMTs) for RRMS have improved patient outcomes over the past 20 years, effective treatments for those entering progressive disease stages are lacking [8]. Fundamental to establishing effective treatments targeted at progressive stages of MS is increasing our understanding of these pathological mechanisms.

This review focuses on compartmentalised inflammation in MS and in particular, what we know about meningeal tertiary lymphoid structures (TLS; also called B cell follicles) which are organised clusters of immune cells, associated with more severe and progressive forms of MS. Meningeal inflammation and TLS could represent an important fluid or imaging marker of disease activity, whose therapeutic abrogation might be necessary to stop the most severe outcomes of disease.

## 2. Compartmentalised Inflammation and Tertiary Lymphoid Structures in Multiple Sclerosis

In early histological studies of post-mortem MS tissue and mouse models of CNS inflammatory demyelination called Experimental Autoimmune Encephalomyelitis (EAE), pockets of immune cells in distinct areas such as perivascular and periventricular spaces were observed [9,10,11]. Extensive inflammatory infiltrates within the leptomeninges and perivascular spaces were described in a cohort of post-mortem MS cases that had active demyelination [9]. In a further study of post-mortem tissue, lymphatic-like channels were identified in perivascular spaces of chronic MS lesions [10]. This lymphatic tissue appeared to have similar features to that of the antibody-producing regions of lymph nodes. Leptomeningeal infiltrates were also observed within the spinal cord of a chronic relapsing EAE mouse model [11].

Over the past couple of decades our knowledge of inflammatory infiltrates within CNS compartments and their relevance in the pathophysiology of MS has continued to expand. Pathological hallmarks of early MS and RRMS are active white matter lesions accompanied by a breakdown of the blood–brain barrier (BBB) [12]. In the later disease stages and in progressive MS, inactive, smoldering (slow expansion of pre-existing demyelinated lesions) or shadow (remyelinated) white matter lesions are most commonly observed, with the findings of active white matter lesions relatively rare [13]. In the progressive disease stages evidence of BBB disruption is also rare and inflammation becomes compartmentalised behind an intact or repaired BBB [14]. Focal areas of inflammation can be observed within the meninges of progressive MS cases [15]. Within the meninges, these clusters of immune cells can exist as simple groupings of B and T cells to more organised cellular aggregates reminiscent of secondary lymphoid organs (SLOs). Secondary lymphoid organs include lymph nodes, spleen, Peyer’s patches, bronchus-associated lymphoid tissue (BALT) and mucosa-associated lymphoid tissue (MALT) and are involved in surveying self and foreign antigens to determine if an adaptive immune response should be initiated [16,17]. When clusters of immune cells gather in ectopic, non-lymphoid sites and demonstrate a more organised assemblage similar to SLOs, they are termed tertiary lymphoid structures [18,19,20]. Within the literature there are interchangeable terms such as ectopic lymphoid organs, ectopic lymphoid tissue, lymphoid-like structures and B cell follicles used to describe these organised cell clusters. For the purpose of this review, we will refer to these structures as tertiary lymphoid structures (TLS).

TLS have been identified in both post-mortem MS cases (Table 1) and mouse models of MS [15,21,22,23,24,25,26,27,28,29,30,31]. In post-mortem tissue, TLS have been identified in the meninges of the cerebral hemispheres, particularly within the deep sulci, cerebellum, brain stem and spinal cord [15,23,25,32,33]. Tertiary lymphoid structures are not unique to MS and have been identified in other immune-mediated conditions, infections and cancers [18]. In some disease processes the presence of TLS is associated with a favourable prognosis and in others, a more severe disease course occurs. In immune-mediated conditions, TLS tend to be found at sites of chronic inflammation and are associated with a more severe disease course [18]. Useful insights have been gained from studies in rheumatoid arthritis (RA) were the identification of TLS within synovial fluid of patients can be used to help inform treatment response and disease prognostication. It was identified in these studies that patients with RA, who had TLS, tended to have a greater disease severity and a poorer response to anti-tumour necrosis family (TNF)-α therapy [34,35].

**Table 1 biomedicines-10-02604-t001:** Summary of human TLS studies.

Study	Cases	TLS Identified	Findings	Location of TLS
Serafini et al., 2004 [25]	SPMS n = 3 PPMS n = 2 RRMS n = 1 Control (non-neurological disease) n = 1 Post-mortem FFPE and fixed frozen tissue	2 SPMS cases had TLS. 0 TLS identified in PPMS and RRMS	TLS consisted of CD3^+^ T cells, CD35^+^ cells, CXCL13^+^ cells, Ki67^+^ nuclei. Plasma cells were identified in periphery of follicle.	In TLS+ cases, inflammatory infiltrates found around blood vessels in cerebral leptomeninges. B cells accumulated perivascularly in chronic inactive lesions and in subarachnoid space.
Magliozzi et al., 2007 [15]	SPMS n = 29 PPMS n = 7 Controls n = 3 Post-mortem FFPE and snap frozen tissue	PPMS: 0 TLS cases, 3 cases demonstrated moderate meningeal inflammation SPMS: TLS cases n = 12	IHC staining identified inflammatory infiltrate of CD3^+^ T cells, CD20^+^ B cells, CD138^+^ or Ig^+^ plasmablasts/plasma cells, CD68^+^ macrophages. TLS had evidence of CD35^+^ and CXCL13^+^ cells, Ki67^+^ B cells and Ig^+^ plasmablasts/plasma cells.	TLS observed in frontal, temporal, parietal lobes and cingulate gyrus. 1 TLS identified in brainstem. TLS always found adjacent to subpial lesions and along depth of cerebral sulci. TLS cases associated were with more severe grey matter pathology.
Serafini et al., 2007 [36]	MS n = 22 Other inflammatory neurological conditions n = 7 Non-neurological controls n = 2 Alzheimers n = 1 Non EBV related lymphoblastic leukaemia n = 1 Post-mortem FFPE and fixed frozen tissue	B follicles consisting of CD20^+^ B cells, clustered around network of stromal/FDCs that expressed CXCL13. Furthermore, contained cells expressing AID and caspase-3. Cells expressing anti-apoptotic molecule bcl-2 also observed.	ISH staining for EBER transcripts on 8 MS cases with B follicles identified EBER^+^ cells within meninges, with maximal enrichment in follicles (n = 15), perivascular cuffs of acute (n = 4) and chronic active (n = 16) WM lesions. 70–90% of EBER^+^ cells identified as CD20^+^ B cells. The highest percentage of B cells expressing EBERs were detected inside and around ectopic B cell follicles suggesting that these structures may originate from the expansion of EBV infected B cells. Ectopic B cell follicles contained numerous LMP1^+^ but no EBNA2^+^ cells. A high frequency of BFRF1^+^ cells was observed inside and around all intrameningeal B cell follicles analysed indicating that these structures represent main sites of viral reactivation. Double immunostainings showed that BFRF1 immunoreactivity was present in a substantial proportion of intrameningeal B cells and plasma cells (30–55%) but was much stronger in plasma cells than in B cells.	B follicles observed in cerebral meninges.
Kooi et al., 2009 [37]	PPMS n = 7 Progressive relapsing n = 1 SPMS n = 12 MS subtype not determined n = 8 Controls n = 6 Post-mortem FFPE tissue	0 TLS cases	Meninges from chronic MS patients contained more CD3^+^ T cells, CD68^+^ macrophages, DC-SIGN dendritic cells than controls. There were fewer CD20^+^ B cells and CD138^+^ plasma cells were seen occasionally in chronic MS meninges versus controls. Meningeal inflammation was not found to be associated with adjacent subpial demyelination.	TLS not observed in this cohort
Willis et al., 2009 [38]	23 FFPE tissue specimens from 12 MS cases with confirmed B cell infiltrates and 12 fixed frozen MS cases with meningeal tissue were examined for EBV by ISH. 17 snap frozen MS lesions with confirmed B cell infiltrate from 5 cases and 12 snap frozen MS tissue containing meninges from 12 cases were examined for EBV by rt-PCR.	B cell aggregates identified in 3/12 cases within brain parenchyma 4/12 cases had a loose B cell infiltrate within the meninges.	In tissue specimens each containing white matter lesions (n = 23), EBV was examined by ISH - all were negative for the EBV transcript EBER. Subset of cases further examined with IHC for expression of EBV latent proteins and found to be negative for LMP1 and EBNA2. RT-PCR was used to detect genomic EBV or EBER1. Neither detected from 17 snap frozen specimens from 5 MS cases. All examined tissue specimens had CD20^+^ B cells detected.	B cell aggregates in brain parenchyma
Frischer et al., 2009 [39]	67 MS Cases - 9 acute (died within 1 year disease onset) - 5 RRMS - 35 SPMS - 13 PPMS - 5 benign - 28 non neurological disease controls FFPE post-mortem tissue	Follicle structures identified in meninges in 15/67 cases	In SPMS and PPMS cases, follicles were only present in cases with active progressive disease. Active demyelination and neurodegeneration was only observed in cases with pronounced inflammation in the brain.	Infiltrates of CD3^+^ T cells, CD20^+^ B cells and Ig^+^ plasma cells identified in meninges. B cells and plasma cells noted to predominantly accumulate in the perivascular spaces and the meninges.
Peferoen et al., 2010 [40]	Screened 632 CNS specimens from 94 MS cases (Netherlands Brain Bank) 12 blocks from 12 cases used in Serafini et al. JEM 2007 (UK MS Society Brain Bank). Post-mortem FFPE tissue used for EBER ISH (as per supplementary material)	Screening of 11 patients (76 blocks) did not identify B follicles as defined by the presence of CXCL13 or podoplanin. 60 blocks from 16 patients had evidence of prominent B cell infiltrates but ectopic/lymphoid follicles not observed. All UK samples negative for EBV encoded RNA technique. IHC for BZLF1, BMRF1, BFRF3 and BLLF1 + LMP1 also performed.	B cell rich areas screened for EBV encoded RNA and EBV viral lytic (BZLF1, BMRF1, BFRF3, BLLF1) and latent (LMP1) proteins. Nuclear EBV encoded RNA was found in only one tissue specimen from a single MS case. All others negative. RT-PCR to search for EBV genomes and encoded RNAs in 5 tissue blocks containing B cell rich areas did not detect EBV DNA or RNA. Single sample (EBV encoded RNA Negative) was positive for multiple EBV lytic cycle markers (BZLF1, BMRF1, BLLF1). All samples screened from UK MS Society Brain Bank cohort were negative for EBV encoded RNA and EBV lytic and latent proteins.	TLS not observed in this cohort
Magliozzi et al., 2010 [22]	SPMS n = 37 Controls (no neurological disease) n = 14 Non-MS neurological inflammatory diseases n = 9 Post-mortem FFPE, snap frozen and fixed frozen.	SPMS: TLS cases n = 20 TB meningitis: TLS cases n = 1 Luetic meningitis: TLS cases n = 1	TLS demonsrtated Ki67^+^CD20^+^ B cells, CD35^+^ and CXCL13^+^ stromal cells/FDCs, Ig^+^ plasmablasts. Lacked GCs.	Frontal, temporal, parietal, occipital lobes examined. CD20^+^ B cells detected along and in depth of some sulci of frontal, temporal, parietal lobes-in particular cingulate and precentral gyrus. TLS always adjacent to subpial lesions.
Torkildsen et al., 2010 [41]	Microarray analysis of 6 MS and 8 controls (no neurological disease) Post-mortem FFPE and snap frozen. For qPCR validation–5 additional MS and 4 controls used.	No CD20^+^ B cell follicles detected in meninges. CD3^+^ cells found in meninges and active lesions. Small number of CD20^+^ B cells in meninges and cortex.	572 probes were identified as differentially expressed between MS samples and controls: 296 downregulated and 276 upregulated in MS. 33 of 276 upregulated genes mapped to molecular function Ig. A total of 83 Ig-related probes showed detectable expression among the MS and control samples, targeting 67 unique genes. Almost half of all Ig-related genes present show a significant upregulation in cortical samples of MS patients compared with controls. qPCR on RNA samples with primers specifically targeting the transcripts of the EBV lytic protein BZLF1 and the latent proteins LMP1 and 2 and EBNA1 and 2 showed no signs of latent or lytic EBV infections in any of the MS or control samples.	CD20^+^ B cell follicles not detected in meninges.
Serafini et al., 2010 [42]	SPMS n = 9 (Post-mortem snap or fixed frozen) Non-neurological controls n = 3 (Post mortem FFPE or snap frozen)	Ectopic B follicles (n = 12) identified in 8 out of 9 MS cases.	In all MS brain specimens analysed, LMP-2A (EBV encoded latent membrane protein) immunoreactivity localised to surface of many lymphocytes within perivascular cuffs of active and chronic active WMLs, meningeal immune infiltrates and B cell follicles. Double immunofluorescence staining showed majority of LMP-2A^+^ cells were CD20^+^ B cells. 80–90% of B cells within meningeal follicles were CD27^+^ antigen experienced cells. 2 MS cases had evidence of BAFF^+^ cells in sparse meningeal infiltrates and B cell follicles. In B cell follicles BAFF^+^ cells co-localised with CD20^+^ B cells and the percentage of B cells expressing BAFF ranged between 10% and 50%. Within the B cell follicles, LMP-2A^+^ cells expressing BAFF ranged between 15% and 90%. BAFF was rarely detected in meningeal iba-1 positive microglia/macrophages.	B cell follicles were detected in meninges.
Howell et al., 2011 [23]	SPMS n = 123 Post-mortem FFPE or fixed frozen.	107/123 (87%) at least one sample with moderate inflammatory meningeal infiltrate. 64/123 (52%) substantial (‘++’) perivascular or meningeal inflammatory infiltrate. 49/123 (40%) had evidence of TLS	TLS+ cases had a 6 fold increase in total grey matter lesion area. There was a greater meningeal inflammatory infiltrate. TLS+ cases presented at earlier ages and entered progressive stages sooner.	TLS predominantly found in deep cerebral sulci–cingulate, insula, temporal and frontal gyri. Always in close association with subpial lesions.
Lucchinetti et al., 2011 [43]	Brain biopsy samples with sufficient cortex available n = 138. FFPE tissue. 43/138 cases had meningeal tissue available. 13/43 cases with meningeal tissue were excluded due to surgical haemorrhage.	TLS not specifically looked for.	15/43 cases with meningeal tissue had evidence of cortical demyelination. Diffuse meningeal inflammation was associated with cortical demyelination, particularly subpial lesions.	IHC staining showed the presence of perivascular meningeal infiltrates containing CD3^+^ T cells and CD20^+^ B cells.
Lovato et al., 2011 [44]	Post-mortem (half of samples FFPE and half snap frozen) 11 MS cases - 7 SPMS - 3 chronic progressive - 1 RRMS CSF obtained from 1 case post-mortem.	Meningeal B cell aggregates identified in a subset of MS cases with CD20 IHC analysis. Number of cases with meningeal B cell aggregates not stated.	This study characterised B cell repertoires from meningeal B cell aggregates and the corresponding parenchymal infiltrates from brain tissue. Observed similar features of antigen experience in B cell clones. The relative clonal expansion of B cells in meningeal aggregates was 24% and in parenchymal infiltrates 28%. Mutation frequency in Immunoglobulin (Ig) variable region heavy chain (VH) sequences similar in both areas. ~90% IgG isotype and remainder IgM. This is in contrast to the IgG/IgM ratio of 15:85 expected in peripheral blood of healthy controls.	B cell aggregates identified in meninges.
Choi et al., 2012 [24]	PPMS n = 26 Controls n = 6 326 frozen blocks (50 fixed and 276 snap frozen) and 416 paraffin-embedded blocks from 26 PPMS cases 24 FFPE blocks from control cases were examined.	0 TLS	8/26 cases had substantial (‘++’) meningeal and perivascular immune cell aggregates. Meningeal inflammation was evident in this cohort of PPMS cases. A greater extent of meningeal inflammation was associated with greater neurite loss and more severe MS disease.	8 cases with substantial meningeal immune cell aggregates had evidence of CD3^+^ T cells and CD20^+^ B cells. Other defining features of TLS; Ki67^+^ and CD20^+^ double positive cells and CD35^+^ cells were not observed in these aggregates.
Magliozzi et al., 2013 [45]	181 brain tissue blocks (cerebral cortex) from 44 SPMS cases - 26 Follicle positive (F+) - 18 Follicle negative (F-) 17 frozen tissue blocks from cerebral hemisphere of 11 follicle positive SPMS cases	F+ cases with infiltrated cortical lesions had CD20^+^ B cells and Ig^+^ plasmablasts/plasma cells. 11/26 subpial cortical lesions contained dense perivascular immune infiltrates. 0/18 F- cases contained infiltrated cortical lesions.	ISH for EBER demonstrated EBER positive cells in 3/3 intracortical perivascular infiltrates analysed. EBER positive cells also detected in adjacent inflamed meninges and in WMLs. EBER signals typically nuclear. Perivascular cells showing nuclear reactivity for BZLF1 (early EBV lytic stage) were detected in all infiltrated active cortical lesions (n = 4) but not in chronic active cortical lesion. BZLF1-positive cells were also detected in the meninges adjacent to the infiltrated cortical lesions. BFRF1 immunoreactivity was expressed in a substantial proportion of Ig^+^ plasmablasts/plasma cells in the cortical perivascular cuffs, the adjacent meninges and in WMLs.	Follicle+ cases had a higher frequency of active WMLs and a higher frequency of active and chronic active cortical lesions. Perivascular immune infiltrates in cortex typically formed around small venules in layers II and III.
Howell et al., 2015 [46]	SPMS n = 27 (Subset of cases previously characterised [23] with known TLS (n = 12) and TLS negative (n = 15) Non-neurological controls n = 11 Post-mortem FFPE	0 TLS in cerebellar blocks examined. One forebrain TLS+ case had evidence of substantial meningeal infiltration (+++) within the cerebellum however unable to further analyse due to loss of the area of interest in subsequent tissue sections.	This study investigated the extent of meningeal inflammation within the cerebellum of a previously characterised cohort of TLS+ and TLS- SPMS cases. Cases that were known to have TLS in the forebrain meninges had evidence of mild-moderate (‘+’ and ‘++’) meningeal immune cell infiltrate within the cerebellum. Inflammation of the meninges was associated with more extensive cerebellar GM demyelination.	TLS not observed within this cohort of cerebellar tissue.
Serafini et al., 2016 [47]	SPMS n = 15 (29 cerebral tissue blocks examined) PFA-fixed frozen snap-frozen sections Controls: fixed-frozen and snap-frozen human adult lymph nodes (1 abdominal and 2 hilo-pulmonary lymph nodes) and 1 snap frozen tonsil	TLS n = 5	TLS contained CD35^+^ stromal cells. RORγt immunoreactivity was mainly nuclear and observed in 6 SPMS cases (5 TLS+ and 1 TLS-). This was almost exclusively localised to meninges. RORγt^+^ cells found in the periphery of 12 out of 18 TLS, generally clustering in areas enriched with CD3^+^ T cells.	Located in meninges.
Bevan et al., 2018 [33]	Short MS disease duration n = 12 Progressive MS n = 21 Non diseased controls n = 11 Other neurological inflammatory disease controls n = 6 Post-mortem FFPE	4 out of 12 cases in short disease duration cohort had evidence of TLS.	TLS+ cases had evidence of increased meningeal CD68^+^ macrophages, CD3^+^ T cells and CD20^+^ B cells. TLS comprised CD8^+^ cytotoxic T cells, B cells co-expressing PCNA and leukocytes expressing transcripts of CXCL13.	TLS were associated with subpial GML. TLS+ cases had extensive neurodegeneration and elevated parenchymal microglia/macrophage activation.
Hassani et al., 2018 [48]	MS n = 101 Non-MS control n = 21 Other neurological controls n = 9 Post-mortem FFPE	Minimal-moderate meningeal infiltration observed. Negligible lymphoid aggregates in examined sections.	This study investigated for the presence of EBV in MS brain tissue detecting EBV by PCR and EBER ISH. 47/101 cases were positive for EBV in meninges. EBER ISH showed EBV positive cells in 83/101 MS cases–80/101 EBV+ cells detected in brain parenchyma and 60/101 EBV+ cells detected in meninges. 5/21 non-MS neurological controls had evidence of EBV+ cells. Double staining IHC performed on 18 EBV heavily infected cases identified 11/18 and 7/18 cases were found to be double positive for EBV/GFAP and EBV/Iba-1 respectively.	
Bell et al., 2019 [49]	11 PPMS 22 SPMS 2 PD 13 healthy controls Post-mortem FFPE	11/22 SPMS cases TLS+ 0/11 PPMS cases had TLS	CD20^+^ B cells, CD35^+^ cells, CD138^+^ plasma cells and CXCR5 expression on lymphocytes detected in TLS. Investigated for evidence of GC function; 4/38 TLS positive for Bcl-6 but all expressed CXCR5 (homing receptor for GCs) Investigated for evidence of regulatory T cells within TLS-FOXP3^+^ cells not detected within follicles and almost completely absent in tissue section.	75% of chronic active lesion brain and 88% of chronic active lesion spinal cord cases had evidence of CD3^+^ T cell and/or CD20^+^ B cell infiltrates.
Reali et al., 2020 [32]	22 SPMS 5 non neurological controls Spinal cord meninges from 11 TLS+ cerebral cases 11 TLS- cerebral cases Post-mortem fixed-frozen.	TLS were observed in 3/11 spinal cord meninges that had known cerebral TLS.	TLS comprised CD20^+^ B cells, CD35^+^ FDCs, CD3^+^ T cells, CD8^+^ T cells, Ki67^+^ B cells and Ig^+^ plasma cells. Cases that had TLS within spinal cord meninges had more CD20^+^ B cells in comparison to TLS- cases (3-fold higher in number in the TLS+ SPMS cases). CD4^+^ T cells were found to be more numerous within meninges, grey and white matter perivascular cuffs in TLS+ cases. There was a trend towards greater areas of demyelinated white and grey matter in TLS+ cases (*p* = 0.052). The density of meningeal B cells in TLS+ SPMS correlated with the extent of axon loss in the lateral corticospinal tract, dorsal column and combined spinal cord tracts axons.	Spinal cord meninges.

Abbreviations: MS, multiple sclerosis; SPMS, secondary progressive MS; PPMS, primary progressive MS; RRMS, relapsing–remitting MS; FFPE, formalin-fixed paraffin embedded; TLS, tertiary lymphoid structure; IHC, immunohistochemistry; EBV, Epstein–Barr virus; AID, activation-induced cytidine deaminase; WM, white matter; EBER, Epstein-Barr encoding region; EBNA, Epstein-Barr nuclear antigen; FDC, follicular dendritic cell; ISH, *in situ* hybridisation; LMP, latency membrane protein; rt-PCR, reverse transcription-polymerase chain reaction; CNS, central nervous system; RNA, ribonucleic acid; DNA, deoxyribonucleic acid; TB, tuberculosis; GC, germinal centre; qPCR, quantitative polymerase chain reaction; WML, white matter lesion; BAFF, B-cell activating factor; GML, grey matter lesion; GM, grey matter; PCNA, proliferating cell nuclear antigen; GFAP, glial fibrillary acidic protein.

In post-mortem MS studies, TLS were initially observed in SPMS cases [15]. More recently however TLS have also been identified in early MS [33,43]. It is difficult to ascertain the incidence of TLS in MS. TLS have been observed in up to 54% of post-mortem SPMS cases however these are selected cohorts [22]. The incidence of TLS in early MS cases is unknown and due to the rarity of short disease duration post-mortem MS cases, will be difficult to establish.

In MS it is often difficult to predict which patients will develop a more active or progressive disease course, particularly at early stages when patients are first facing treatment decisions. Furthermore, predicting patient response to DMTs is challenging. Whilst it is not feasible to study TLS in vivo in humans at this point, we can gain an understanding of TLS composition, formation and function from mouse models of MS and post-mortem studies. A better understanding of TLS in MS will be essential to inform studies investigating biomarkers and therapeutic targets in progressive MS and aggressive disease courses. Given that the meninges appear to be the primary site of TLS in MS it is important to gain an understanding of how immune cells traffic within the meninges and enter/exit to peripheral and CNS compartments.

### 2.1. The Meninges; Vascular Channels and Lymphatic Networks

The meninges consist of three layers; the dura, arachnoid and pia mater, that envelope the cortex. The subarachnoid space lies between the arachnoid and pial layers and contains cerebrospinal fluid (CSF). Produced by the choroid plexus, CSF circulates through the ventricular system and subarachnoid space and has important functions for mechanical protection of the CNS and cerebral autoregulation. The blood-CSF barrier also has an important role in the transport of immune cells and molecules, and is responsible for the removal of waste products from the CNS [50].

The CNS was once thought to be an immune-privileged site however, we now know that immune cells can traffic within the meninges and through the CSF, providing surveillance for foreign antigens. Indeed the meninges harbour a diverse range of immune cells with distinct populations between the brain parenchyma, lymphatic system and even between the different meningeal layers, providing an important interface between the periphery and CNS parenchyma [51,52,53,54]. The meninges can therefore serve as a site for coordinating adaptive immune responses and orchestrating homeostatic and neuroinflammatory responses [55].

The dura and leptomeninges (arachnoid and pia mater) comprise distinct vascular compartments. The dura mater consists of vast arterial, venous and capillary networks, which lack tight junctions and includes large venous sinuses (the sagittal and transverse sinuses), which drain blood from the cerebral veins into the systemic circulation [54]. The pial vascular network has connections into the CNS parenchyma. Unlike the dural vasculature, pial blood vessels have tight junctions, selectively transporting molecules and immune cells into and out of the CSF and parenchyma [54].

The CNS was generally described as devoid of lymphatic vessels however CNS lymphatics were described in the late 18th century [56]. There has been renewed interest in the CNS lymphatic network with several recent studies identifying meningeal lymphatic vessels within the dura [57,58]. This dural lymphatic network appears to be an important site for the movement of molecules, immune cells, interstitial fluid and CSF from the CNS to the periphery [57,59]. Evidence from experimental animal models suggest that the dural sinuses are a critical site for immune surveillance. In the healthy state, CNS antigens from the CSF have been found to gather in the dural sinuses where antigen presenting cells interact with circulating T cells [60]. Furthermore T cells appear to be able to traffic from the circulation into the meningeal lymphatic vessels and then drain into the deep cervical lymph nodes (cLN) [57,58,61].

Interestingly, in a mouse model of EAE, removing the deep cLNs was associated with a reduction in EAE severity [62]. In a recent study, again using a mouse model of EAE, meningeal T cells were observed to drain to deep cLNs in a CCR7-dependent manner [59]. Furthermore this study observed that when the meningeal lymphatic system was reduced, this resulted in a reduced inflammatory response and less severe pathology [59]. These studies suggest that CNS draining lymph nodes could be a site for antigen presentation and lymphocyte activation and that the meninges are an important site for this inflammatory process.

In a study investigating immune cell trafficking into the CNS, a mouse model was used to label bone marrow-derived cells (neutrophils) in the skull and tibia [63]. This study identified distinct vascular channels within the skull bone and found these to be directly connected to the vasculature of the dura mater [63]. Furthermore, when CNS inflammation was induced in a mouse model of ischaemic stroke and aseptic meningoencephalitis, neutrophils originating from the skull bone marrow, migrated to the inflamed sites and were far more numerous than neutrophils originating from the tibial bone marrow in both these mouse models. Interestingly the authors also imaged portions of skull bones from 3 patients that underwent craniectomy and identified similar channels within the inner skull cortex connecting to marrow cavities [63].

By characterising the phenotype of meningeal B cells in young and aged mice, single cell RNA sequencing (scRNA-seq) and B cell receptor sequencing (BCR-seq) data identified that a large proportion of dural B cells were in early stages of B cell development [64]. This was in contrast to blood, where very few early B cell markers were identified making it unlikely that dural B cells were derived from the systemic circulation [64]. The authors of this study also found evidence of vascular channels within the skull and using a CD19 reporter mouse, identified that IgM^−^ B cells migrated through these vascular channels from the skull bone marrow to the meninges [64].

These studies challenge previously accepted concepts about immune cell migration in the CNS and suggest that perhaps cellular migration and recruitment to inflamed sites, is occurring independent of the systemic circulation. Understanding the meningeal vascular and lymphatic networks and their connections to the periphery will likely give us important clues about immune cell trafficking and potential sites of CNS antigen presentation to T cells. However, our understanding of immune processes in these networks is limited in MS and further studies will be required if we are to understand this important aspect of immunoreactivity in diseased states.

Understanding how cells and molecules traffic to and from the meningeal compartment will be important for our ability to detect and monitor the meningeal inflammatory microenvironment in MS. Increasing our knowledge of the vascular and lymphatic networks connecting through the meninges and CNS parenchyma and how these relate to, or reflect, the peripheral lymphatics and vasculature, will be important in our search for serum or CSF biomarkers for disease prognostication. 

### 2.2. Composition of Tertiary Lymphoid Structures

Most of our initial understanding of the composition, formation and functional mechanisms of TLS arose from studies investigating SLOs [18,19,20,65,66,67]. Tertiary lymphoid structures are often defined by features of SLOs [18]. Secondary lymphoid organs consist of distinct B and T cell zones, germinal centres (GCs), follicular dendritic cells (FDCs), reticular networks comprising fibroblastic reticular cells (FRCs) and high endothelial venules (HEVs).

The composition of TLS in other immune-mediated diseases and cancers often show similarities to that of SLOs [18,65]. In contrast, the composition of cellular clusters in MS can range from aggregates of B cells, aggregates of B and T cells, to more organised structures reflective of SLOs (Figure 1). The more organised structures are often defined as TLS in MS if there is evidence of B cells, T cells, FDCs and cell proliferation within the cluster [23]. However, post-mortem studies have shown heterogeneity of cellular organisation within TLS in MS and this heterogeneity has also been demonstrated in several mouse models of EAE [15,21,28].

**Figure 1 biomedicines-10-02604-f001:**
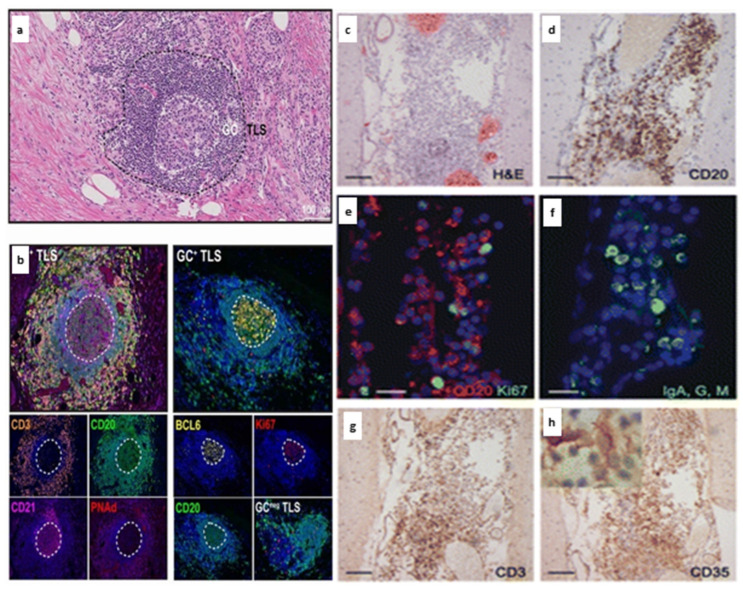
Tertiary lymphoid structures in cancer and multiple sclerosis [23,68]. Tertiary lymphoid structures (TLS) in pancreatic cancer demonstrate features reminiscent of secondary lymphoid organs with evidence of germinal centres (**a**), distinct B and T cell zones and PNAd^+^ high endothelial venules (**b**). BCL6^+^ cells indicate germinal centre reactions and cells undergoing proliferation (Ki67^+^) are concentrated in the germinal centre (**b**). In comparison, meningeal TLS in MS typically demonstrate a loose aggregate of CD20^+^ B cells (**d**) interspersed with proliferating (CD20^+^Ki67^+^) B cells (**e**), immunoglobulin A, G, M^+^ plasma cells/plasmablasts (**f**), CD3^+^ T cells (**g**) and CD35^+^ follicular dendritic cells (**h**). Permissions: (**a**,**b**) Gunderson AJ et al. (2021) Germinal center reactions in tertiary lymphoid structures associate with neoantigen burden, humoral immunity and long-term survivorship in pancreatic cancer, OncoImmunology, 10:1, DOI: 10.1080/2162402X.2021.1900635 by permission of Taylor & Francis Ltd. www.tandfonline.com. Accessed on 18 July 2022. (**c**–**h**) Howell OW et al. Meningeal inflammation is widespread and linked to cortical pathology in multiple sclerosis. Brain 2011 134; 2755–2771 by permission of Oxford University Press. Accessed on 5 July 2022.

In EAE, with disease progression into chronic stages, B cell aggregates show evidence of developing into more organised structures reminiscent of SLOs [30]. In post-mortem SPMS studies TLS can contain CD3^+^ T cells, CD20^+^ B cells, CD138^+^ plasma cells and CD68^+^ macrophages within the meninges [25]. Follicular dendritic cells play an important role in the cellular organisation and B cell recruitment into TLS. The prominent B cell chemoattractant CXCL13 is produced by FDCs and binds to the CXCR5 receptor expressed on a subset of T helper cells [69,70,71]. In post-mortem SPMS cases, FDCs expressing CXCL13 within TLS have been observed [25].

B cell activating factor (BAFF), a member of the TNF family, also has a critical role in B cell development, proliferation and survival [72,73]. In SJL mice, levels of CXCL13 and BAFF mRNA expression were elevated in relapsing–remitting and chronic-relapsing EAE [21]. This was accompanied by the formation of organised lymphoid structures within the meninges, some of which contained CD35^+^ and M1-expressing FDCs [21]. In further studies using SJL mice and MBP-PLP fusion protein (MP4)-induced EAE, FDCs have also been observed and appear to co-localise with B cell aggregates [28,30]. In human MS studies, BAFF mRNA levels in monocytes and BAFF-receptor mRNA levels in B and T cells have been found to be increased in comparison to healthy controls [74]. An upregulation of BAFF expression in both acute and chronic MS lesions has also been observed in a post-mortem study [75]. In this study, BAFF expression co-localised with astrocytes suggesting they were the pre-dominant cell type producing BAFF and highlights the key role of resident cells in coordinating recruitment and retention of lymphocytes to the CSF-filled spaces. 

Another feature of SLOs are reticular fibres which provide a structural collagen support network to trap and traffic lymphocytes [65]. In two studies using SJL mice immunised with PLP_139-151_, FRCs and the formation of reticular networks within the meninges was observed [21,31]. Reticular networks have also been observed in TLS in other mouse models of EAE, sometimes encapsulating the entire immune cell aggregate [29,30]. These reticular networks are not always observed or as well formed in TLS in post-mortem cases [25].

To facilitate the trafficking of lymphocytes, SLOs develop HEVs. These vascular networks are not a frequent finding within TLS observed in EAE. In a mouse model of MP4-induced EAE, HEVs have been observed however this is not a consistent finding in other mouse models [29,30]. In a cohort of post-mortem SPMS cases, HEVs were not observed within TLS [25]. Perhaps the good vascular networks that exist within the meningeal compartment or differing adhesion molecules expressed on meningeal blood vessels may explain this lack of HEVs.

Another important feature of SLOs is the presence of GCs. Within SLOs, GCs are the sites in which B cells undergo somatic hypermutation and clonal expansion to produce memory B cells and plasma cells [18]. In other immune-mediated conditions, GCs have been observed within TLS and appear to maintain the autoimmune response [76,77]. Both in mouse models of EAE and post-mortem MS cases, proliferating B cells and plasma cells have been observed, suggesting the presence of GCs [21,25,29]. Whilst there is evidence of some germinal centre features within TLS in post-mortem MS cases, this is not a conclusive finding across all observed TLS [25]. 

### 2.3. What Drives the Formation of Tertiary Lymphoid Structures?

The formation of TLS appears to share similarities to that of SLOs [65,78,79,80]. SLO development involves members of the TNF-lymphotoxin (LT) family [17,65]. Interactions between CD45^+^CD4^+^CD3^−^ lymphoid-tissue initiator (LTi) cells-a group 3 innate lymphoid cell (ILC) population, and tissue resident stromal organiser cells results in the production of cytokines and chemokines involved in B and T cell recruitment [16,17,65,79,80]. The LT heterotrimeric complex, LTα_1_β_2_, is expressed on the surface of embryonic LTi cells, B cells, natural killer cells and activated T cells. LTα_1_β_2_ binds to the LTβ receptor (LTβR) which is expressed on stromal cells, dendritic cells and macrophages. This binding induces the expression of chemokines such as CCL19, CCL21 and CXCL13, which recruit and organise T and B cells within TLS [65,80,81]. The interaction between LTi cells and stromal organiser cells also promotes the development of HEVs, which facilitate lymphocyte trafficking in SLOs [17,79,81].

The initiating cell(s) and signalling pathways for TLS development in MS are unclear. Mouse models investigating the formation of TLS in EAE indicate that Th17 cells may play a significant role in the initiation of TLS. Following the adoptive transfer of Th17 cells into C57BL/6 mice, TLS formation was observed in 16 of 22 recipients during EAE [29]. In contrast, TLS formation was not observed following the adoptive transfer of Th2 and Th9 cells, and in only one of seven Th1 cell-recipient mice, despite development of disease in the majority of cases. This suggests Th17 cells are playing a pivotal role in TLS formation. Further evidence that Th17 cells have an initiating role in TLS development was demonstrated in a MP4-induced EAE mouse model [82]. In this mouse model, B cell aggregate formation is observed. The study found that there was an absence of CD3^−^CD5^−^CD4^+^RORγt^+^ LTi cells however CD3^+^CD5^+^CD4^+^RORγt^+^ Th17 cells were observed in the cerebellum of mice with EAE. Th17 cells were observed at peak of disease and again during the chronic stages but this time at higher frequencies. Th17 cells were also observed in close association with TLS formation in another study using a MP4-induced EAE mouse model, although the functional role of Th17 cells was not investigated in this study [30].

In a study of post-mortem tissue from SPMS patients, there was also a suggestion that Th17 cells and group 3 ILCs play a role in TLS formation [47]. The authors identified retinoic acid receptor-related orphan receptor-γt (RORγt)^+^ cells within, or in close proximity to, the majority of identified B-cell aggregates. Most of the RORγt^+^ cells co-expressed CD3 and a minority were CD3 negative, suggesting these cells could represent Th17 cells and group 3 ILCs respectively.

Similarities exist between Th17 and LTi cells. Both cells express RORγt and LTα_1_β_2_ and produce IL-17 and IL-22 [83,84]. For the initiation of TLS in MS, Th17 cells may adopt a similar role to that of the LTi cell in SLO formation. IL-27, which inhibits Th17 cells and EAE [85,86,87,88], has also been shown to inhibit TLS formation in a mouse model of inflammatory arthritis [89]. It would be interesting to determine whether a similar inhibitory effect of IL-27 on TLS development would be observed in a mouse model of EAE harbouring TLS.

The LT signalling pathway also appears to be implicated in TLS formation in EAE. The SJL-proteolipid protein_139-151_ (PLP_139-151_)-immunised mouse model develops a relapsing–remitting clinical disease course somewhat similar to RRMS. A study using this model demonstrated an upregulation of LTβ and LTβR gene expression in the CNS at EAE onset and during subsequent relapses [28]. In addition, the authors found that blocking of the LTβR resulted in a reduced inflammatory infiltrate, reduced B cell aggregation within the meninges and TLS were not observed in those cases. Further evidence that LTβR signalling supports the accumulation of B cells within TLS was demonstrated in another EAE study using SJL mice [31]. Furthermore, the authors reported that following the adoptive transfer of Th17 cells, LTβ expression was required to propagate inflammation and disease. 

Germinal centre-derived memory B cells are the primary site of latent Epstein–Barr virus (EBV) infection [90,91]. There is evidence that TLS in MS can be a reservoir of latent EBV with some studies showing that meningeal B cell follicles can harbour latent EBV transcripts (EBV-encoded small nuclear mRNA) and EBV latent membrane protein LMP-2A [36,42]. The persistence of latent EBV in meningeal B cells could be a factor in the propagation and expansion of TLS in MS. This is interesting particularly given the recent study from Bjornevik et al. demonstrating EBV as a clear risk factor for the development of MS, with a 32-fold increase in MS risk following EBV seroconversion [92].

The factors required for TLS formation and ongoing maintenance are complex and likely involve multiple cell signalling pathways. Investigating mechanisms of TLS formation in human MS tissue is challenging, however if we can gain a better understanding of TLS cellular composition, particularly during different disease phases, we may gain better insights of potential immune signals required for TLS development and maintenance. 

### 2.4. Potential Function of Tertiary Lymphoid Structures

#### 2.4.1. Could TLS Be a Site for B Cell Maturation and Immunoglobulin Synthesis?

The presence of oligoclonal bands (OCBs) in CSF is a common feature in MS, being present in over 95% of cases [93]. The detection of OCBs is included as part of the diagnostic criteria for MS [94]. Alongside the presence of OCBs, the findings of somatically hypermutated and clonally expanded B cells in the CSF of MS patients suggest there is an antigen-driven immune component to disease pathogenesis [95,96,97,98,99]. Whether these B cells undergo expansion and maturation within the meningeal compartment or are recruited into the CNS from the periphery, or potentially both, is unclear.

Germinal centres within SLOs such as the spleen and lymph nodes, are sites of B cell maturation and production of antibody-secreting plasma cells in the periphery. In response to the presentation of antigen, B cells within the GC proliferate, initially within the dark zone where they undergo somatic hypermutation. GC B cells differentiate into centroblasts and centroctyes. Proliferating centroblasts are involved in immunoglobulin class switching and during this process express activation-induced cytidine deaminase (AICD). This leads to affinity maturation and the differentiation of GC B cells into memory B cells and long-lived plasma cells [100,101,102].

In autoimmune diseases such as Sjogren’s syndrome (SS), rheumatoid arthritis, Hashiomoto’s thyroiditis (HT) and Graves’ disease (GD), TLS exhibiting GCs are thought to contribute to the production of autoantibodies. Tertiary lymphoid structures demonstrating cardinal features of GCs with distinct dark and light zones, HEVs, dendritic cells and mantle zones have been identified in thyroid glands of patients with HT and GD [77]. Higher levels of serum autoantibodies, anti-thyroperoxidase (TPO) in HT and anti-thyroid stimulating hormone receptor (TSH-R) in GD are associated with patients who have TLS. Similarly in SS, patients who have TLS in salivary glands (the affected organ in SS) have been found to have significantly higher levels of anti-Ro and anti-La autoantibodies [103]. This suggests that GCs within TLS are playing a role in the production of pathogenic autoantibodies. Further evidence of GC function is the identification of populations of clonally expanded B cells and hypermutation of the immunoglobulin V genes in salivary glands in SS [104] and the expression of AICD in TLS found in synovial fluid of rheumatoid arthritis patients [105] indicating somatic hypermutation of Ig genes is occurring. Plasma cells producing the pathogenic antibody anti-citrullinated protein (ACPA) have been identified in close proximity to TLS in RA further implicating TLS GCs as a source of autoantigen [105].

Could meningeal TLS in MS therefore serve as a site for the production of clonally expanded B cells and IgG synthesis? In the CSF of MS patients, immunoglobulins are often class-switched and have signs of affinity maturation which would suggest that GC reactions are occurring within the CNS compartment [95,98,106]. In some post-mortem SPMS cases, TLS have demonstrated features of GCs comprising FDCs, proliferating B cells and plasma cells [25] however it is unclear whether these TLS demonstrate all typical features of GCs including the distinct dark and light zones and further studies investigating this are needed. In the post-mortem MS TLS studies reviewed here (Table 1), the presence of CSF OCBs was not available in the included data. Given CSF OCBs are detected in >90% of MS patients, it is unlikely that OCB production is dependent on TLS however there could be a distinct pattern of OCB antibodies in TLS+ cases which may provide important clues on disease pathogenesis.

Investigating the functional role of TLS in MS is difficult in humans. Animal models of MS may help us to address key mechanistic questions of GC formation and function. In some mouse models of MS, lymphoid aggregates with evidence of GCs have been identified. The proliferation marker Ki67, CD138^+^ plasma cells and AICD have been observed in TLS in MP4-induced EAE [30]. One study did not observe GC-like reactions in early EAE although difficulty with CD138 staining may have led to an underestimation [27]. More recently, in a MOG-induced EAE rat model, lymphoid structures closely resembling SLOs with distinct T and B cell zones and HEVs were observed in the meninges [107]. These animal models may help advance our understanding of how TLS form, are maintained, their functional role(s) and how they are potentially affected by treatments. However, it is important to note the limitations of EAE models. The most commonly used model of MOG_35-55_-induced EAE usually features acute onset of ascending clinical signs from tail tip to limbs caused by a high number of concurrent, rapidly developing spinal cord, optic nerve and brain lesions. There is little clinical recovery in this model with persistent neurological impairment from the initial acute events. It is also important to note that EAE studies routinely use female, young adult rodents with limited follow-up time relative to the life-long nature of MS after diagnosis which affects both sexes. Thus, notwithstanding the considerably usefulness of EAE model in the development of current DMTs for MS, the models fall short of representing the full spectrum of neuropathological processes in MS.

Whether meningeal TLS are the sites GC reactions and production of antibody secreting plasma cells in MS remains to be determined. It will also be important to establish if GCs can form within TLS in early disease stages or if are they restricted to TLS in a more chronic inflammatory environment. The lack of capacity to detect or study TLS in living patients is a key limitation in understanding the pathological importance of these structures and developing an approach to tackle this limitation would be a major advance for both clinical and research purposes.

#### 2.4.2. Could TLS Propagate Subpial Neuroinflammation and Contribute to Progression in MS?

Evidence from post-mortem studies demonstrates that cases with TLS had a more severe MS disease course, entering progressive stages sooner and acquiring disability at faster rates [25,28,32]. A hallmark of the progressive stage of MS is the accumulating neurodegeneration and cortical demyelinating pathology, which is related to the extent of overlying inflammation nested in the meninges [22,23,108]. Inflammatory cortical demyelination is also observed in early MS and correlates with a worse disease outcome [43].

We also know from post-mortem studies, that cases with TLS have a greater degree of inflammatory infiltrate within the meninges, which correlates with a larger number and area of cortical demyelinating lesions [15,24,25,25]. TLS are often, but not always, found overlying subpial cortical lesions and there is a relative grade of inflammation extending from the subpial surface to the subcortical white matter, suggesting TLS are involved in the propagation of neuroinflammation [15].

In post-mortem MS tissue, higher degrees of meningeal inflammation are also associated with increased gene and protein expression of TNF-α and interferon (IFN)-γ. These expression levels are further increased in cases that have TLS [109]. Furthermore in post-mortem TLS cases there is upregulation of TNF receptor 1 (TNFR1) and genes involved in the TNF/TNFR1 necroptotic signalling pathway and downregulation of caspase-8 dependent apoptosis [110,111]. These studies indicate that TNF signalling is being directed towards TNFR1 necroptotic cell death and that this is potentially being influenced by cytokine release from meningeal inflammatory cells. In a rat model of chronic meningeal inflammation, introduction of TNF-α and IFN-γ into the CSF resulted in meningeal inflammation and cortical neurodegeneration, increased neuronal expression of TNFR1 and activation of necroptotic signalling pathways [111,112]. This demonstrates that in this animal model, meningeal inflammation and a cytotoxic milieu can lead directly to subpial demyelination and substantial neurodegeneration in a pattern similar to that observed in progressive MS.

#### 2.4.3. Identifying which Patients Have Higher Degrees of Meningeal Inflammation and a High Likelihood of TLS

Identifying which patients have higher degrees of meningeal inflammation and TLS could help predict disease course, inform therapeutic strategies and assess response to DMTs. The difficulty is how we identify such patients. Given the close proximity of the pial meningeal layer to the subarachnoid space, investigating CSF immune profiles could provide important clues for the meningeal inflammatory microenvironment.

Indeed, there does appear to be distinct CSF profiles in MS cases. In a study seeking to identify CSF biomarkers of intrathecal inflammation, CSF profiles of MS, other inflammatory neurological disease (OIND) and non-inflammatory neurological disease (NIND) were investigated [113]. Raised CSF levels of IL-8 and IL-12p40 were found to distinguish MS cases from the OIND group, furthermore CSF IL-12p40 levels were distinct between the PPMS and RRMS groups. This study highlighted that distinct CSF profiles exist between neurological inflammatory diseases and between different subtypes of MS. An investigation of CSF profiles in post-mortem MS cases with known TLS, reported an increased expression of inflammatory cytokines such as IFN-γ and TNF-α and chemokines/cytokines related to lymphoid neogenesis, such as CXCL13, CXCL10, IL-6 and IL-10 [114]. In a clinical study investigating CSF profiles and magnetic resonance imaging (MRI) in a cohort of treatment-naïve RRMS patients, after a four year follow up period it was found that patients who experienced evidence of disease activity had higher CSF levels of CXCL13, CXCL12, IFN-γ, TNF, sCD163, LIGHT and APRIL at time of diagnosis [115]. Furthermore, there was a strong correlation between CXCL13 CSF levels and the development of new cortical lesions on MRI, indicating that CSF profiles could be useful to distinguish patients at greater risk of cortical lesion accumulation. The immune signatures of TLS and inflammatory profiles in CSF could therefore potentially predict more progressive and severe forms of MS if found at elevated levels in early disease.

MRI is the gold standard imaging technique to aid diagnosis in MS and specific radiological criteria are included in the McDonald 2017 diagnostic criteria [68,94]. The detection and monitoring of white matter brain and spinal cord MS lesions with MRI can inform treatments decisions and help assess response to DMTs [116]. It is difficult to detect grey matter lesions (GMLs) using conventional MRI techniques routinely available in the clinic. Given the association between extent of grey matter demyelination and progression in MS and the findings of TLS overlying subpial GM, the ability to detect GMLs on MRI could help stratify patients and monitor progressive phases of MS non-invasively.

In recent years, advances in MRI techniques and ultra high field 7 T MRI has improved our ability to detect GMLs and in particular subpial demyelination [117,118,119,120,121,122]. Several MRI studies have investigated the presence of leptomeningeal contrast enhancement (LMCE) as a potential marker for leptomeningeal inflammation in MS. On 3 T and 7 T MRI, LMCE has been identified in cohorts of RRMS, SPMS and PPMS cases and appears to correlate with subpial demyelination and overlying areas of meningeal inflammatory cells [123,124,125,126] but not TLS per se. The highest prevalence of LMCE appears to be in progressive MS were it has been detected in 33–85.7% of cases on 3 T MRI [123,126]. In their study using 3 T MRI, Zivadinov et al. reported a persistence of LMCE in 50% of cases (after a 5 year follow up period) and multiple LMCE foci were more likely to be observed in SPMS [126]. However, findings from a 7 T MRI study suggested a different prevalence and persistence pattern of LMCE with no difference observed between RRMS and progressive MS groups and LMCE persisted in ~70–85% of patients after 2 years [127], although a shorter follow up period and using more sensitive MRI may account for these differences. The presence of LMCE is associated with increased GM volume loss, a faster rate of GM atrophy, longer disease duration and increased disability in MS [123,124,125,126]. Some studies suggest that LMCE is not affected by the use of DMTs [123,126,127,128,129]. In two studies using anti-CD20 therapies, the administration of intrathecal rituximab and intravenous ocrelizumab did not reduce the appearance of LMCE [130,131]. In summary, these studies suggest that LMCE may reflect inflammation and disturbance to the blood-CSF barriers within the meningeal compartment, particularly in progressive cases where chronic inflammation tends to persist within the leptomeninges and MRI could be used to monitor treatment response or lack of.

However, caution does still need to be applied in the interpretation of LMCE as this can be seen in other neuroinflammatory, infective and neoplastic conditions [93]. A further caveat is that most neurology clinics only have access to 1.5 T or 3 T MRI scanners where the detection of LMCE is difficult. At present the utility of 7 T MRI is largely confined to research. The increasing use of advanced MRI techniques within clinical trial settings however will hopefully further expand our ability to detect radiological features of GMLs and LMCE and eventually pave a way for routine clinical use. Alongside CSF analysis, the use of MRI will be vital to help better predict which patients are more likely to enter progressive disease stages and assess treatment response.

## 3. Therapeutic Strategies Targeting Meningeal Inflammation and TLS

Alongside observational studies, evidence from clinical trials has shown that anti-CD20, B cell depleting therapies, reduce relapse rates in patients with RRMS and slow disease progression in PPMS and SPMS [132,133,134,135]. Given that B cells are a predominant cell type within TLS and that TLS associate with more severe and progressive forms of MS, could targeting these structures with B cell therapies, slow or halt progression?

In a mouse model of EAE that develops immune cell aggregates reminiscent of TLS, anti-CD20 treatment was administered prior to EAE onset and after EAE onset to investigate whether anti-CD20 treatment could prevent formation of TLS or change the disease course. Treatment with anti-CD20 resulted in depletion of B cells within the peripheral blood and lymph nodes, however meningeal lymphoid structures still developed. Furthermore, the authors observed that administering anti-CD20 treatment after EAE onset did not affect the clinical course of spontaneous chronic EAE. Anti-CD20 treatment however did influence B cell composition of TLS, with reduced numbers of B220^+^ B cells and increased numbers of MPO^+^ neutrophils [136].

Similarly in the opticospinal encephalomyelitis (OSE) mouse, anti-CD20 treatment administered after disease onset depleted B cells peripherally, however B cell aggregates were still observed in spinal cord meninges, and no effect was observed on disease progression [137]. The effects of anti-CD19 CAR-T cell administration was investigated and it was found that whilst there was reduction in TLS formation, paradoxically, the progressive disease course was exacerbated. These studies suggest that depleting peripheral B cell populations alone will be insufficient to reduce or stop TLS formation and may not alter the progressive disease course. These studies highlight our lack of understanding of TLS and the need for caution when directly or indirectly targeting these structures with therapeutic interventions.

Interestingly the DMT siponimod, which is licenced for SPMS, was found to reduce the formation of meningeal lymphoid structures in a mouse model of EAE [138]. It was observed that when siponimod was administered prior to EAE onset, formation of TLS was reduced. Similarly, when siponimod was administered at peak of disease, there was a reduction in TLS in comparison to the vehicle treated control mice. Siponimod is a sphingosine 1-phosphate receptor (S1PR) 1,5 modulator which traps B and T lymphocytes within lymph nodes, thus reducing both B and T cells in the peripheral circulation. Siponimod is also known to cross the BBB and therefore may also have a direct effect on meningeal inflammation [139]. Future post-mortem studies of tissue from individuals that have been treated with siponimod and B cell-depleting DMTs may provide insight to the effect of these interventions on either the development, or the persistence, of these structures in the CNS. 

Another interesting therapeutic target is Bruton’s tyrosine kinase (BTK). BTK is an enzyme of the B cell receptor signalling pathway, which if inhibited, reduces B cell proliferation [140]. Multiple BTK inhibitors have been developed and are currently in clinical use for a number of immune-mediated and lymphoproliferative disorders [141]. In vivo studies suggest BTK inhibitors reduce meningeal inflammation. In a mouse model which demonstrates meningeal inflammation, MRI with post-contrast imaging observed that treatment with the BTK inhibitor evobrutinib, reduced meningeal contrast enhancement in comparison to controls [142]. In RRMS patients, a placebo-controlled phase 2 trial of evobrutinib demonstrated that patients receiving evobrutinib had fewer enhancing lesions on MRI [143]. Whilst no significant difference was observed in annualised relapse rate (ARR) or expanded disability status scale (EDSS), there was a relatively short follow-up period. BTK inhibitors therefore could have therapeutic potential in progressive MS, potentially via targeting B cells outside and as well as within the CNS, and further clinical trial data are eagerly awaited.

## 4. Conclusions

There is clear evidence that patients who have TLS have a more severe MS disease course, however our understanding of the functional role of TLS and the reason for the heterogeneity observed in meningeal inflammation is lacking. Given the heterogeneity of TLS in MS, future research should investigate the cellular composition of TLS, exploring B and T cell subset analysis in the different disease stages. Increasing our understanding of the composition of TLS and determining if there are distinctions in the immune profiles of TLS in different disease stages, may provide further insights into how these structures develop, how they could be contributing towards chronic inflammation in MS and how they could be targeted therapeutically. Investigating a post-mortem cohort treated with MS DMTs, with different mechanisms of actions, such as natalizumab (selective binding of α_4_β_1_-integrin), ocrelizumab, ofatumumab and rituximab (anti-CD20 monoclonal antibodies), fingolimod and siponimod could also reveal further insights into TLS formation and potential alterations that may occur to TLS structures following treatment.

Whilst our knowledge of inflammation within the meningeal compartment and TLS in MS has increased dramatically over the past two decades, it is important to continue advancing our knowledge of TLS and the contribution TLS have on progressive pathogenic mechanisms. If we can better understand TLS immune signatures and bioactivity, we can inform future studies searching for prognostic markers and the development of therapeutic targets that will slow or halt progression in MS.

## Data Availability

Not applicable.

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
