# Peer review of "A Review of Compartmentalised Inflammation and Tertiary Lymphoid Structures in the Pathophysiology of Multiple Sclerosis"

_biomedicines, 2022, doi:10.3390/biomedicines10102604_

Round 1

Reviewer 1 Report (Previous Reviewer 2)

I thank the Authors for the inclusion of the MRI aspects in the manuscript.

Reviewer 2 Report (Previous Reviewer 1)

All raised points are addressed adequately.

This manuscript is a resubmission of an earlier submission. The following is a list of the peer review reports and author responses from that submission.

Round 1

Reviewer 1 Report

In the presented review Kee and colleagues investigated the potential role of meningeal tertiary lymphoid structures (TLS) often also referred to as B cell follicles. This is a well-conducted review which provides an important summary of the compartmentalised inflammation in MS. It is well structured and the discussion and conclusions are reasonable. I only have some minor points to be addressed by the authors:

1. As the authors state TLS can mainly or solely be investigated in post-mortem tissue or mouse models. Can the authors therefore please discuss the potential influence of lifelong therapy and disease duration on TLS analysis in post-mortem tissue and how this is affected by growing therapy options as in a few years there most likely won't be any post-mortem tissue of an untreated patient which might limit the insights into real pathological processes. And also other limitations of the EAE model especially for progressive MS types need to be discussed in brief.

2. Can the authors give an outlook on how those limitations of TLS analysis might be overcome in the future (e.g. 3D tissue culture, advanced imaging techniques, etc.)

3. The intrathecal production of antibodies in the CNS is one major hallmark of MS and the presence of oligoclonal IgG bands in the CSF is part of the diagnostic criteria of MS. The current concept of OCB production is based on long-living plasmablasts and B-cell follicles in the meninges. Can the authors please discuss the role of TLS in local antibody production and if there are data avialable compare the presence of TLS to OCB production

Reviewer 2 Report

The manuscript by Kee et al. describes the immunopathology of meningeal tertiary lymphoid structures in multiple sclerosis. This narrative review is very well written and is a great resource for many readers of Biomedicines. My main comment for this review is the lack of translational data that could further supplement the pathophysiological/histology principles that are well outlined in the manuscript.

I personally disagree with the paragraph outlined in line 380-386. There is a plethora of in vivo studies in MS patients that characterize and describe the relationship between leptomeningeal follicles and clinical, MRI and serum outcomes. The manuscript will be significantly improved by include additional section on imaging of leptomeningeal contrast enhancement (LM CE), an outcome indicative of the TLS. I am going to outline only several manuscripts on this topic, given the extensive literature.

Imaging of LM CE in MS models:

-          Pol S, Schweser F, Bertolino N, Preda M, Sveinsson M, Sudyn M, Babek N, Zivadinov R. Characterization of leptomeningeal inflammation in rodent experimental autoimmune encephalomyelitis (EAE) model of multiple sclerosis. Exp Neurol. 2019 Apr;314:82-90. doi: 10.1016/j.expneurol.2019.01.013. Epub 2019 Jan 23. PMID: 30684521.

Examples of references in humans:

-          Zurawski J, Lassmann H, Bakshi R. Use of Magnetic Resonance Imaging to Visualize Leptomeningeal Inflammation in Patients With Multiple Sclerosis: A Review. JAMA Neurol. 2017 Jan 1;74(1):100-109. doi: 10.1001/jamaneurol.2016.4237. PMID: 27893883.

-          Hildesheim FE, Ramasamy DP, Bergsland N, Jakimovski D, Dwyer MG, Hojnacki D, Lizarraga AA, Kolb C, Eckert S, Weinstock-Guttman B, Zivadinov R. Leptomeningeal, dura mater and meningeal vessel wall enhancements in multiple sclerosis. Mult Scler Relat Disord. 2021 Jan;47:102653. doi: 10.1016/j.msard.2020.102653. Epub 2020 Dec 4. PMID: 33333417.

-          Absinta M, Cortese IC, Vuolo L, Nair G, de Alwis MP, Ohayon J, Meani A, Martinelli V, Scotti R, Falini A, Smith BR, Nath A, Jacobson S, Filippi M, Reich DS. Leptomeningeal gadolinium enhancement across the spectrum of chronic neuroinflammatory diseases. Neurology. 2017 Apr 11;88(15):1439-1444. doi: 10.1212/WNL.0000000000003820. Epub 2017 Mar 10. PMID: 28283598; PMCID: PMC5386437.

Detailed meta-analysis seen here:

-          Ineichen BV, Tsagkas C, Absinta M, Reich DS. Leptomeningeal enhancement in multiple sclerosis and other neurological diseases: A systematic review and Meta-Analysis. Neuroimage Clin. 2022;33:102939. doi: 10.1016/j.nicl.2022.102939. Epub 2022 Jan 10. PMID: 35026625; PMCID: PMC8760523.

Effect of disease modifying therapies on LM CE:

-          Bhargava P, Wicken C, Smith MD, Strowd RE, Cortese I, Reich DS, Calabresi PA, Mowry EM. Trial of intrathecal rituximab in progressive multiple sclerosis patients with evidence of leptomeningeal contrast enhancement. Mult Scler Relat Disord. 2019 May;30:136-140. doi: 10.1016/j.msard.2019.02.013. Epub 2019 Feb 11. PMID: 30771580; PMCID: PMC7325522.

-          Zivadinov R, Jakimovski D, Ramanathan M, et al. Effect of ocrelizumab on leptomeningeal inflammation and humoral response to Epstein-Barr virus in multiple sclerosis. A pilot study. Multiple Sclerosis and Related Disorders 2022;67.